# Highly stacked 3D organic integrated circuits with via-hole-less multilevel metal interconnects

Hocheon Yoo[1,4], Hongkeun Park [2,4], Seunghyun Yoo[1], Sungmin On[1], Hyejeong Seong[3], Sung Gap Im[2] & Jae-Joon Kim[1]

Multilevel metal interconnects are crucial for the development of large-scale organic integrated circuits. In particular, three-dimensional integrated circuits require a large number of vertical interconnects between layers. Here, we present a novel multilevel metal interconnect scheme that involves solvent-free patterning of insulator layers to form an interconnecting area that ensures a reliable electrical connection between two metals in different layers. Using a highly reliable interconnect method, the highest stacked organic transistors to date, a three-dimensional organic integrated circuits consisting of 5 transistors and 20 metal layers, is successfully fabricated in a solvent-free manner. All transistors exhibit outstanding device characteristics, including a high on/off current ratio of ~$10^7$, no hysteresis behavior, and excellent device-to-device uniformity. We also demonstrate two vertically-stacked complementary inverter circuits that use transistors on 4 different floors. All circuits show superb inverter characteristics with a 100% output voltage swing and gain up to 35 V per V.

[1] Department of Creative IT Engineering, Pohang University of Science and Technology (POSTECH), Pohang 790-784, Korea. [2] Department of Chemical and Biomolecular Engineering, Korea Advanced Institute of Science and Technology (KAIST), 291 Daehak-ro, Yuseong-gu 34141, Korea. [3] Department of Materials, Imperial College London, London SW7 2BP, UK. [4]These authors contributed equally: Hocheon Yoo, Hongkeun Park. Correspondence and requests for materials should be addressed to S.G.I. (email: sgim@kaist.ac.kr) or to J.-J.K. (email: jaejoon@postech.ac.kr)

Organic thin-film transistors (OTFTs) have been extensively investigated for use in functional electronic device applications, such as wearable electronics[1,2], artificial skins[3,4], and flexible sensors[5–7]. Such applications are typically realized by integrating unit OTFTs into integrated circuits (ICs) on a large scale. However, reducing the size of OTFTs for large-scale integration is not easy due to their strong susceptibility to high temperatures[8,9] and solvent-based developers[10,11], which are essential components of lithography-based high-resolution patterning. Another promising idea that has attracted research interest for large-scale integration of organic electronic devices is 3D stacking of OTFTs because more transistors can be vertically integrated in the same area without requiring a reduction in the transistor size[12–17]. Unlike conventional silicon or metal-oxide semiconductors, organic semiconductors have a low process temperature (~200 °C), thus providing a sufficiently low thermal budget for stacking multiple transistors.

However, using conventional via-hole processes based on lithography for 3D integration of OTFTs is not simple because conventional interconnect techniques with etching-based via-hole processes[18–20] require the use of solvent-based developers or exposure to plasma, which can significantly damage organic semiconductors and drastically degrade the IC performance. Alternative approaches such as laser drilling[3,15] or solvent-based inkjet printing[21,22] have also been used to form via-holes using a laser or by directly injecting solvent on targeted spots, respectively. The organic semiconductor and polymer insulator can be removed in a localized, selective way without degradation in the transistor area and followed by filling the patterned hole with a conductive material to form a via-hole, which creates an electrical connection between two metals in different layers. However, many challenges still exist with these approaches. First, the etching procedure with high-energy laser irradiation is inevitably accompanied by a substantial increase in temperature on the substrate, which may damage materials that are vulnerable to high temperature[23]. This thermal effect may be even greater when the wavelength of the laser beam does not well match the absorption behavior of the dielectric materials being drilled[24]. Additionally, depending on the material properties of the dielectric layer, the laser-drilling process may result in a poor edge geometry (such as a polyimide), causing device failure and poor yield[3]. In the solvent-based printing method, only dielectric materials that are soluble in the printing solvent can be used. However, many excellent insulators do not dissolve in common solvents[25,26], which significantly limits material selection. Moreover, the chemical etchant must be removed by an additional flushing step to complete the via-hole. In addition to the difficulties in selecting an orthogonal flushing solvent that does not damage the organic materials underneath the target dielectric layer, the isotropic nature of the etching profile requires precise control of the etching time to achieve a sharp etch stop. Furthermore, approaches to make via-holes may not be suitable for 3D stacking and integration. As devices are vertically stacked, the total thickness of the intermetal dielectric layers increases, and as a result, forming via-holes through the layers becomes difficult. According to a previous study[21], 5–7 inkjet-printing drops are needed to etch 500-nm-thick poly(vinylphenol) (PVP); thus, tens of drops must be injected to etch a μm-thick dielectric layer. In addition, manufacturing procedures based on laser drilling or solvent-based inkjet printing may become extremely time consuming, especially as the number of required via-holes increases, because of the inherently sequential nature of via processing.

For these reasons, approaches for reliable, large-scale fabrication of organic 3D-ICs must overcome many difficulties. To date, despite significant research efforts on 3D integration of organic transistors, the number of multimetal layers has been limited to only 4[15], and the number of vertically stacked transistors has not been higher than 2[12–17]. Therefore, it is crucial to develop a novel via-formation approach compatible with organic semiconductors and capable of providing a high yield and high throughput. To achieve a highly stacked 3D-OTFT beyond current capabilities, a simple, vertical interconnect fabrication process must be developed to support a large number of vertical layers and targeted interconnects at a low process temperature with a robust dielectric patterning method capable of depositing a dielectric layer without degrading organic semiconductor layers.

Here, we present a novel metal interconnect scheme for simple, reliable 3D integration of OTFT circuits. In contrast to conventional approaches that locally remove insulators for via-hole formation, the proposed scheme selectively forms an insulator by coupling a shadow mask with a solvent-free insulator deposition process. A metal layer can be interconnected to another metal layer through the insulator-free, open area. In particular, a number of vertical interconnects can be formed at the same time regardless of the number of target interconnects because the intermetal dielectric layers can be deposited on multiple selected locations in parallel. To demonstrate the proposed interconnect scheme, we designed and tested multilevel metal interconnect structures using an ultrathin (~50 nm), patterned dielectric polymer synthesized by a solvent-free, vapor-phase polymer deposition process, initiated chemical vapor deposition (iCVD)[27,28]. We also determined the layout design rules for reliable interconnections and precise isolation of the top and bottom metal electrodes by characterizing the electrical resistances of various test patterns with the dielectric layers and metal lines. Using the established 3D multilayer stacking scheme, we also demonstrated ultrahigh stacking of OTFTs, i.e., up to 5 transistors per unit area (25 layers per unit area). Two N,N'-ditridecylperylenediimide (PTCDI-C13)-based n-type transistors[29] and three dinaphtho[2,3-b:2′,3′-f]thieno[3,2-b] thiophene (DNTT)-based p-type transistors[30,31] were fabricated on the bottom two floors (1–2F) and top three floors (3–5F), respectively. By using the proposed solvent-free interconnect scheme, we can simply and reliably integrate OTFTs on multilayers without additional via-hole processes.

## Results

**Multilevel metal interconnect structure.** To investigate the feasibility of the proposed interconnect scheme, we first fabricated a multimetal interconnect test structure based on 5 metal layers and 4 dielectric layers of a poly(1,3,5-trimethyl-1,3,5-trivinyl cyclo-trisiloxane) (PV3D3) film deposited by the iCVD process. As reported previously, PV3D3 is a highly crosslinked organosilicon-based polymer and is known as an excellent ultrathin organic dielectric layer with a wide bandgap (~8.25 eV) and relatively low dielectric constant ($k \sim 2.2$). The dielectric layer exhibited an extremely low leakage current of less than $10^{-8}\,A\,cm^{-2}$ up to $5\,MV\,cm^{-1}$, even at a thickness of ~10 nm[32]. Due to its outstanding thermal stability, the PV3D3 layer can withstand a high annealing temperature up to 250 °C without any degradation of its insulating property[33]. The solvent-free deposition process temperature is near room temperature (40 °C), which minimizes potential damage to layers[34]. Rather than locally removing insulators (Fig. 1a), an ultrathin PV3D3 layer is placed between two metal layers, which require isolation, and the metal layer pair is interconnected in the area where the PV3D3 layer is not deposited (Fig. 1b, c). Depending on the PV3D3 layer, the metal lines can be isolated or interconnected to form a more complex structure (Fig. 1d–g, Supplementary Fig. 1). The measurement results show that the two separate metal interconnect paths ($V_A$ to $V_{A\_out}$ and $V_B$ to $V_{B\_out}$) were successfully isolated from each other, but robust 5-level (M1–M2–M3–M4–M5)

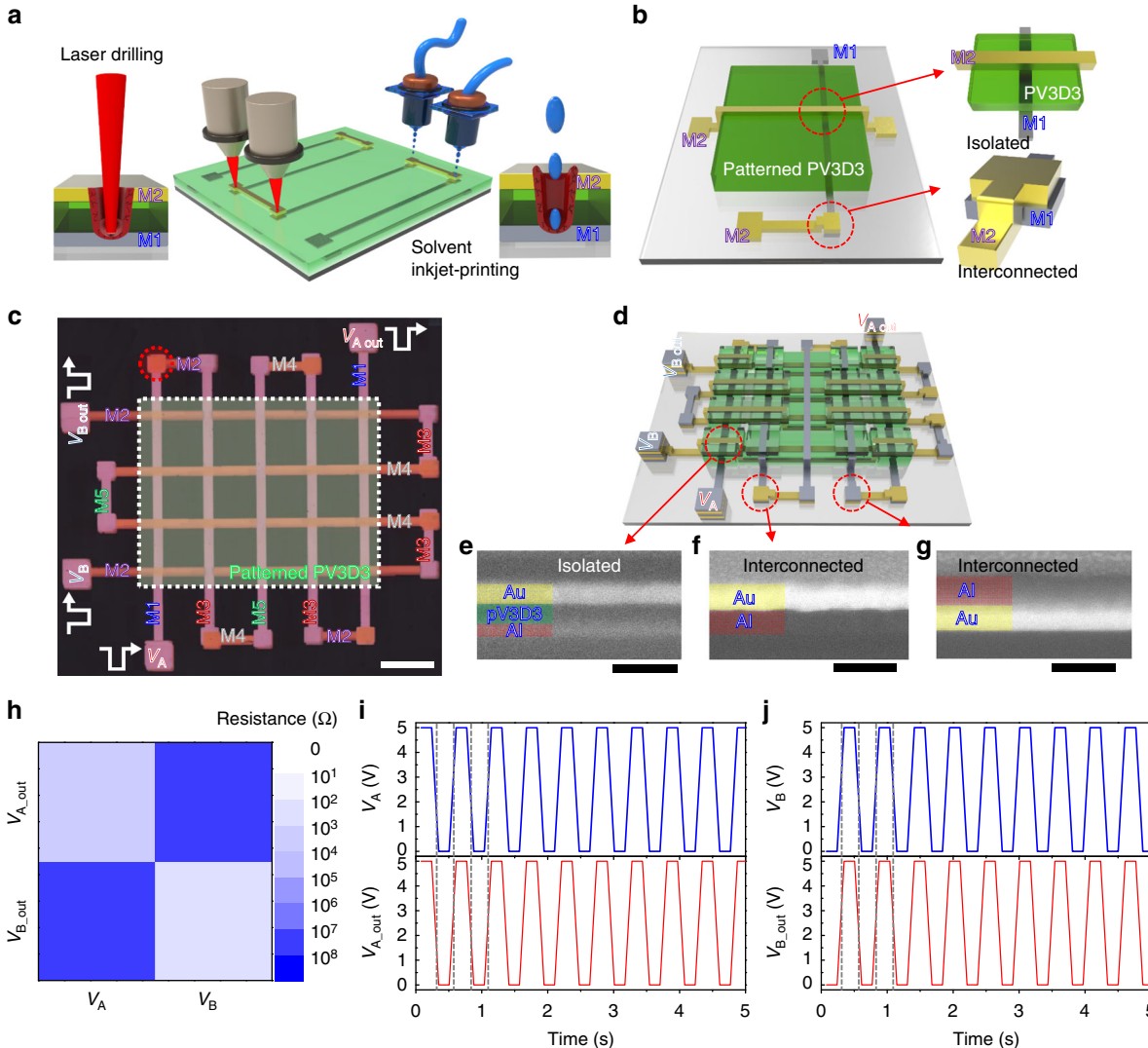

**Fig. 1** Multilevel metal interconnect scheme. **a** Schematic of metal interconnect methods using laser drilling and solvent inkjet printing. **b** Schematic of the metal interconnect method using patterned PV3D3 layers. **c** An optical microscopy image of the fabricated multilevel metal interconnect test structure. The scale bar indicates 1200 μm. **d** Schematic of the multilevel metal interconnect test structure. Cross-sectional SEM images of **e** an isolated metal electrode (M1 and M2), scale bar, 100 nm, **f** an interconnected metal electrode (M3 and M4), scale bar, 100 nm, and **g** an interconnected metal electrode (M2 and M3) scale bar, 100 nm. **h** A mapping image of the resistance values among $V_A$, $V_{A\_out}$, $V_B$, and $V_{B\_out}$. **i** Output voltage response ($V_{A\_out}$) with respect to the input voltage pulse ($V_A$). **j** Output voltage response ($V_{B\_out}$) with respect to the input voltage pulse ($V_B$)

interconnections formed for each path. We measured the resistance between $V_A$ and $V_{A\_out}$ and $V_A$ and $V_{B\_out}$ (Fig. 1h). As intended, the connected line between $V_A$ and $V_{A\_out}$ exhibited a low resistance (~0.91 kΩ), while the isolated path between $V_A$ and $V_{B\_out}$ created by the patterned PV3D3 layers exhibited a high resistance (56 MΩ). We also measured the transient results by simultaneously applying input voltage pulses with opposite signals at $V_A$ and $V_B$ (Fig. 1i, j). The signals were transmitted to $V_{A\_out}$ and $V_{B\_out}$, respectively, without an electrical short between the two terminals. The test results confirmed that separate multilevel metal interconnects were successfully formed using the proposed scheme.

**Layout design rules of via-hole-less multilevel metal interconnect.** As preparation for the demonstration of 3D-OTFT circuits using the proposed interconnect scheme, we evaluated the layout design rules for creating reliable connections and isolating the top and bottom metal lines by characterizing the electrical resistance using various test patterns with the dielectric layers and

metal lines. First, we designed and fabricated Au electrodes on stepped patterns to evaluate high-step coverage of the metal line (Fig. 2a–d, Supplementary Fig. 2). On the patterned PV3D3 layers ($t_{PV3D3} \leq 8$ μm), the lateral metal lines exhibited a significantly lower resistance (~50 Ω) than the baseline resistance ($5 \times 10^8$ Ω), which represents a disconnected case (Fig. 2g). The baseline resistance was measured using a metal-insulator-metal (MIM) structure in which the metal electrodes were separated by a PV3D3 layer ($t_{PV3D3} = 10$ nm). The results showed that a lateral metal connection can be reliably formed over the patterned PV3D3 layer, which has a thickness as high as 8 μm. Next, we evaluated the margin for the overhang of PV3D3 over the metals ($d_{ov}$) for the isolation of the bottom and top electrodes (Fig. 2b–e). When $d_{ov} > 50$ μm, the bottom and top metal layers were completely isolated by the middle PV3D3 layer, resulting in a very high resistance value between the electrodes ($\sim 1.5 \times 10^8$ Ω) (Fig. 2h). We also evaluated the minimum lateral distance ($d_{gap}$) between the patterned PV3D3 layer and the electrodes for reliable vertical interconnection of the electrodes (Fig. 2c–f).

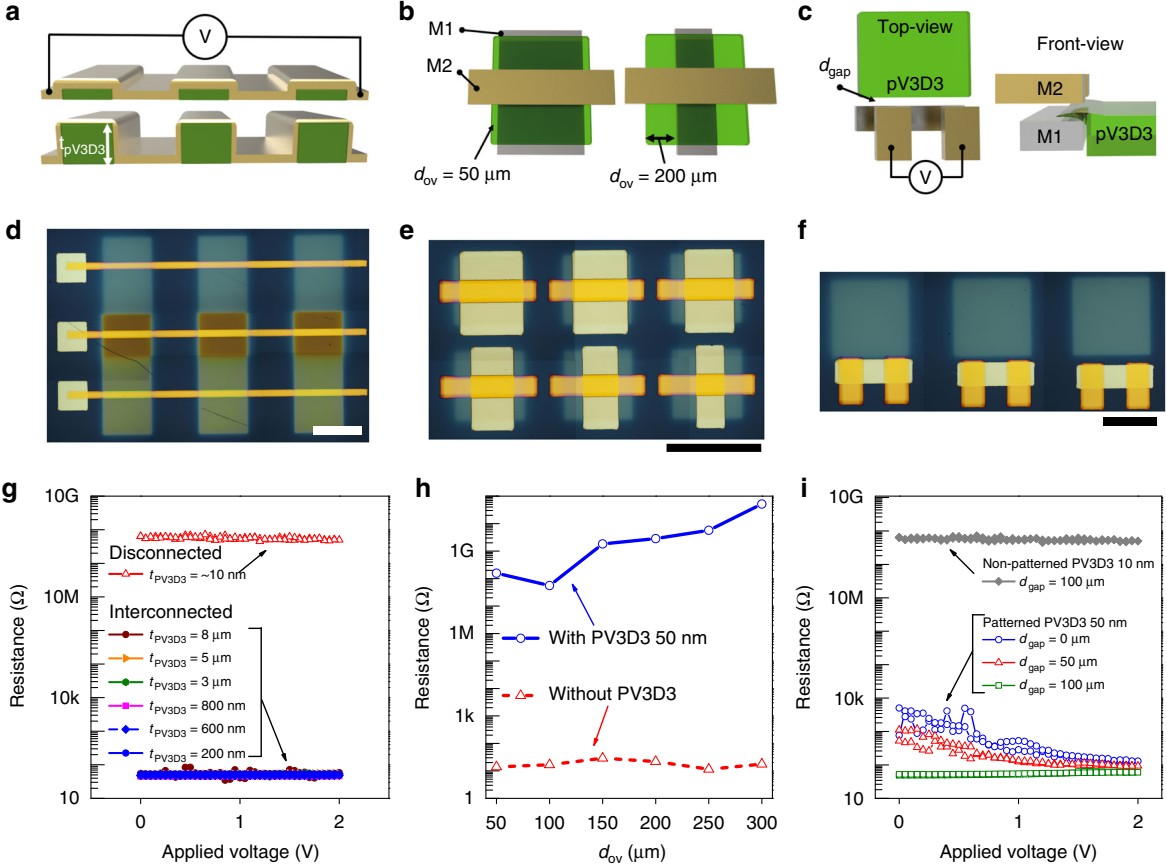

**Fig. 2** Electrical evaluation of metal interconnect and isolation. **a**, **d**, **g** Electrical short test when the electrode is located on a patterned PV3D3 layer as a function of the PV3D3 layer thickness ($t_{PV3D3}$). Schematic (**a**), optical micrograph, scale bar, 1,000 μm, (**d**), and resistance test results (**g**). **b**, **e**, **h** Electrical isolation test when the PV3D3 layer was sandwiched between two metal electrodes as a function of the length of the PV3D3 layer over the electrode ($d_{ov}$). Schematic (**b**), optical micrograph, scale bar, 1000 μm, (**e**), and resistance test results (**h**). **c**, **f**, **i** Electrical interconnect test when the patterned PV3D3 layer was located next to vertically stacked metal layers. Schematic (**c**), optical micrograph, scale bar, 1000 μm, (**f**), and resistance test results (**i**)

When $d_{gap} \geq 100$ μm, the bottom and top electrodes were reliably connected without PV3D3 residue between the electrodes, resulting in a low resistance value between the electrodes (~ 50 Ω) (Fig. 2i).

Based on the results, we suggest the following layout design rules for the proposed scheme:

$$t_{PV3D3} \leq 8 \text{ μm}, \ d_{ov} \geq 50 \text{ μm}, \text{ and } d_{gap} \geq 100 \text{ μm} \quad (1)$$

The capacitance ($C_i$) value of the patterned PV3D3 dielectric layer was $44.7 \pm 2.1$ nF cm$^{-2}$, which was measured at 9 different spots in the MIM structure in an area of 2.1 by 2.1 mm$^2$ (Supplementary Fig. 3), and this value shows the high uniformity of the dielectric layer.

**Vertically-stacked organic transistors**. Based on the proposed interconnect scheme and layout design rules, we designed and fabricated vertically stacked OTFTs (3D-OTFTs). The fabricated 3D-OTFTs consisted of five Al gate electrodes, five Au contact electrodes, two PTCDI-C13 n-type semiconductor layers, three DNTT p-type semiconductor layers, five PV3D3 gate dielectric layers, and four PV3D3 intermetal dielectric layers, which formed two n-OTFTs and three p-OTFTs (Fig. 3a) on the 5 floors in the OTFT stack. The devices were surrounded by gate, source, and drain pads (Fig. 3b–e, Supplementary Fig. 4). At the center of the 3D-stacked OTFTs, the patterned intermetal dielectric layers (~1 μm, PV3D3) were interposed between the transistors (Fig. 3c). The PV3D3 layers were not located at the edges, which is where the multimetal vertical interconnects formed (Fig. 3d).

The PTCDI-C13 required an annealing process ($T_A = 200$ °C) for the formation of edge-on-orientation crystallites, which offer better electrical characteristics[35,36]. The dependence of the crystalline structure of PTCDI-C13 on the annealing temperature ($T_A$) is shown in Supplementary Fig. 5. To anneal only PTCDI-C13 films at $T_A = 200$ °C, we fabricated two n-type PTCDI-C13 OTFTs on the first and second floors (1–2F) and then fabricated three p-type DNTT OTFTs on the third, fourth, and fifth floors (3–5F). A more detailed manufacturing process and optical image of each step for the 3D-stacked OTFTs is shown in Supplementary Figs. 6, 7.

The $I_D$−$V_G$ transfer characteristics were measured at both $|V_D|$ = 2 V and 10 V to investigate the transport properties in both the linear and saturation regions, respectively (Fig. 3f–j). All OTFTs exhibited no hysteresis over the whole range of the applied gate voltage bias ($V_G$). Due to the low leakage current of the PV3D3 dielectric layer, all transistors also showed well-defined off-state regions and high on/off ratios (~10$^7$). The device parameters, such as the on-current ($I_{ON}$), saturation and linear mobility ($\mu_{sat}$ and $\mu_{lin}$, respectively), and threshold voltage ($V_{TH}$), for all transistors were extracted, and the values were comparable to the values reported by others for both p- and n-type transistors regardless of the floor location (Fig. 3k–n). In addition, good linear characteristics were observed in the transfer curves (that is, $I_D^{0.5}$−$V_G$ and $I_D$−$V_G$ dependences in the saturation and linear regions, respectively) for all transistors (Supplementary Figs. 8, 9)[37–39]. To investigate the nonlinearity of the transfer curves, which originated from extrinsic factors that affect charge transport, the reliability factors ($r_{sat}$ and $r_{lin}$) of each

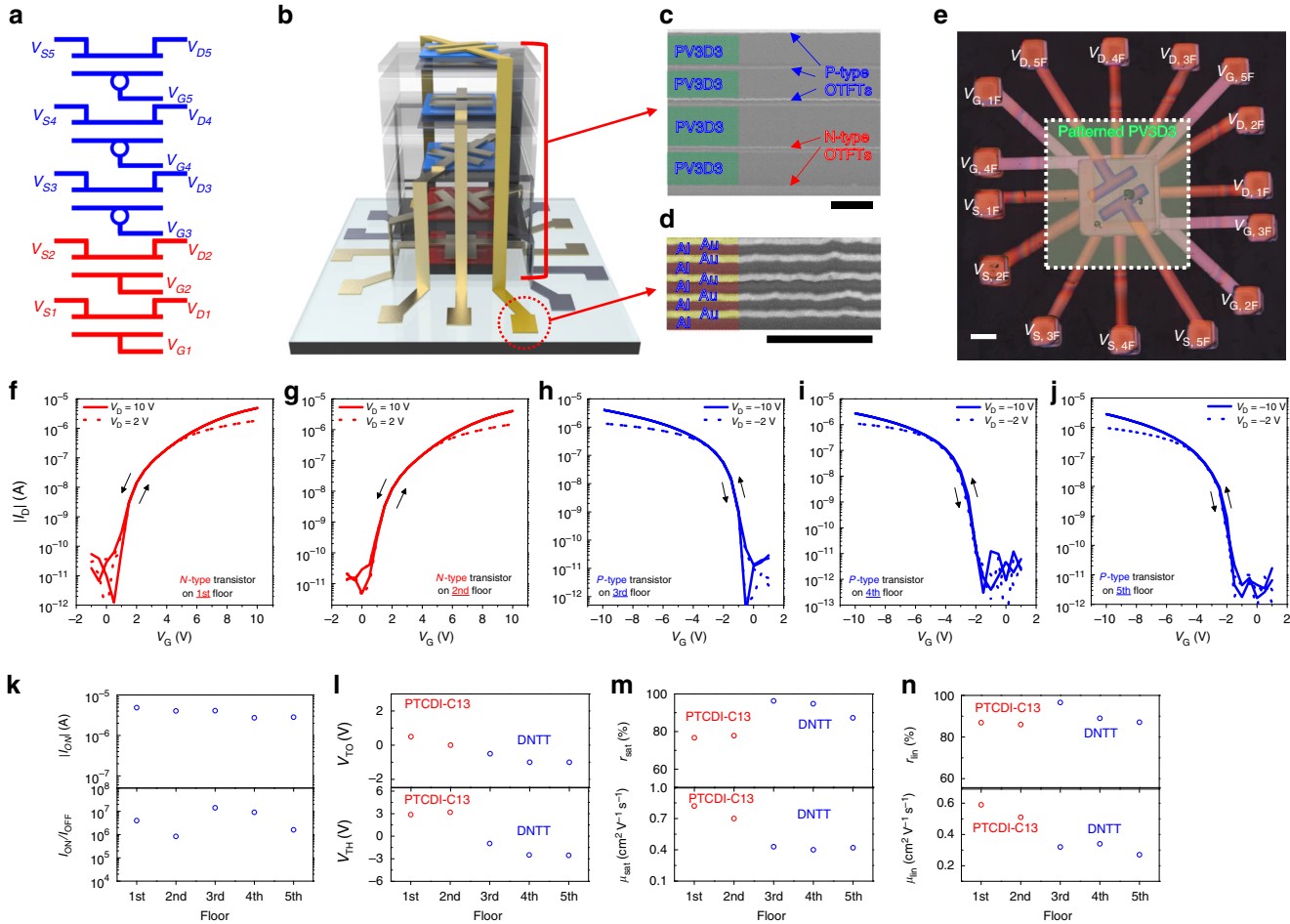

**Fig. 3** 5-layer 3D-stacked OTFT. **a** Schematic of a 5-layer 3D-stacked OTFT. **b** Schematic of a 5-layer 3D-stacked OTFT. Cross-sectional SEM images of **c** a 3D-stacked OTFT, scale bar, 1 μm, and **d** interconnected metal electrodes, scale bar, 500 nm. **e** An optical micrograph image of a 5-layer 3D-stacked OTFT, scale bar, 400 μm. Transfer characteristics of the 3D-stacked **f** PTCDI-C13 n-type transistor at 1F, **g** PTCDI-C13 n-type transistor at 2F, **h** DNTT p-type transistor at 3 F, **i** DNTT p-type transistor at 4F, **j** DNTT p-type transistor at 5F. **k** On/off current ratio and on-current values at $|V_G| = 10$ V. **l** Threshold and turn-on voltage ($V_{ON}$) values. The extracted carrier mobility and reliability factor values **m** in the saturation regime and **n** the linear regime

OTFT were extracted (Fig. 3m, n)[38]. The reliability factor values for the linear and saturation regions simply represent how the transfer curves correspond with the ideal curves and were calculated using Equations (2) and (3), respectively[38]. $I_D^{V_{TH}}$ denotes the drain current at $V_G = V_{TH}$.

$$r_{lin} = \left( \frac{|I_D|^{max} - \left| I_D^{V_{TH}} \right|}{|V_G|^{max}} \right) \bigg/ \left( \left| \frac{\partial I_D}{\partial V_G} \right| \right) \quad (2)$$

$$r_{sat} = \left( \frac{\sqrt{|I_D|^{max}} - \sqrt{\left| I_D^{V_{TH}} \right|}}{|V_G|^{max}} \right)^2 \bigg/ \left( \frac{\partial \sqrt{|I_D|}}{\partial V_G} \right)^2 \quad (3)$$

The purpose of $r_{sat}$ and $r_{lin}$ is to evaluate whether OTFTs have a linear increase in conductivity as the carrier density in the channel increases. Due to contact resistance or charge trapping, OTFTs suffer from nonlinearity, i.e., a 'hump' or 'kink', in their $I_D - V_G$ characteristics[37–39]. In the 3D-OTFTs, all transistors have $r_{sat}$ and $r_{lin}$ values higher than 75%, clearly showing the high quality of the fabricated OTFTs according to guidelines from a previous study[38]. Subthreshold swing (SS) values of less than 450 mV per dec and a relatively low interfacial trap density ($N_t$) on the order of $10^{12}$ cm$^{-2}$ were obtained for the transistors

(Supplementary Fig. 10, Supplementary Note 1), and these values resulted in the hysteresis-free transfer characteristics. The output characteristics are provided in Supplementary Fig. 11.

We also investigated the air-stability of the 3D-OTFTs. To implement more stable device and circuit in air ambient, we encapsulated the 3D-stacked organic transistors with 10 nm-thick Al$_2$O$_3$ layer via atomic layer deposition (ALD) process. Both n-type and p-type 3D-stacked OTFTs were measured at in an inert N$_2$ atmosphere first, and then stored and measured in air ambient (20 °C, 45% relative humidity). To investigate the effect of the ultrathin Al$_2$O$_3$ encapsulation on the device stability, the 3D-stacked devices without the encapsulation layer were also characterized as the negative control (Supplementary Fig. 12a). For the Al$_2$O$_3$-encapsulated 3D-stacked OTFTs, the initial device characteristics were fully maintained throughout the entire measurement period of ~214 h (Supplementary Fig. 12b–d). For the devices without encapsulation, on the other hand, the PTCDI-C13-based n-type transistors showed substantial mobility decrease and $V_{TH}$ shift with the increased air exposure time (Supplementary Fig. 12c). The DNTT-based p-type transistors also showed quite huge $V_{TH}$ shift along with the increased air exposure time (Supplementary Fig. 12d), which is ascribed to the oxygen and moisture in previous reports[40,41]. In the Al$_2$O$_3$-encapsulated 3D-stacked OTFTs, the 10 nm Al$_2$O$_3$ layer was sufficiently good for the protection of the 3D stack devices from

the exposure of oxygen and moisture, leading to the high air-stability result. To further evaluate the air-stability of the $Al_2O_3$-encapsulated 3D-stack OTFTs, we also measured the air-stability of the $Al_2O_3$-encapsulated 3D-stacked inverter as a function of the air exposure time (Supplementary Fig. 13a, b), where the voltage transfer curve of the inverter circuit remained practically identical, demonstrating its excellent environmental stability.

To study the reproducibility and uniformity of the proposed 3D-OTFTs, we reproduced four 3D-stacked buildings including 20 OTFTs. A total of 20 OTFTs were measured for statistical investigation. To verify the switching operation in both linear and saturation regions, the transistors were measured at $|V_D| = 2\,V$ and $10\,V$, respectively (Supplementary Fig. 14). All OTFTs exhibited transistor operation with excellent device performance including high on/off current ratio $\sim 10^5$ and 100% yield. Furthermore, no $I-V$ hysteresis was observed in all the transfer characteristics. For PTCDI-C13-based n-OTFTs, the average saturation and linear mobility values were $\mu_{sat} = 0.56 \pm 0.048\,cm^2\,V^{-1}\,s^{-1}$ and $\mu_{lin} = 0.50 \pm 0.054\,cm^2\,V^{-1}\,s^{-1}$, respectively (Supplementary Fig. 15a–c). For DNTT-based p-OTFTs, the average saturation and linear mobility values were $\mu_{sat} = 0.55 \pm 0.077\,cm^2\,V^{-1}\,s^{-1}$ and $\mu_{lin} = 0.27 \pm 0.055\,cm^2\,V^{-1}\,s^{-1}$, respectively.

**Vertically-stacked complementary integrated circuits**. As another proof-of-concept demonstration, complementary 3D-OTFT inverter circuits (Fig. 4a, Supplementary Fig. 16) were also fabricated. Using the same unit channel width and length ($W = 1000\,\mu m$, $L = 200\,\mu m$), we stacked two PTCDI-C13-based n-type transistors on 1F and 2F and two DNTT-based p-type transistors on 3F and 4F. Gate electrodes ($V_{G,1F}$ and $V_{G,3F}$) for an inverter input ($V_{IN1}$) were vertically connected to each other through an open area in the patterned PV3D3 layers (Fig. 4b). The metal interconnect between the drain electrodes ($V_{D,1F}$ and $V_{D,3F}$) was formed through another open area in the PV3D3 layers to create the inverter output ($V_{OUT1}$). Similarly, we designed a second inverter using an n-type transistor on 2F and a p-type transistor on 4F (Fig. 4c). A more detailed manufacturing process and optical image of each step for the 3D-stacked inverter circuits is shown in Supplementary Figs. 16–19.

The performance of 3D-stacked inverter circuits (1&3F-Inverter and 2&4F-Inverter) was investigated. The $V_{OUT}-V_{IN}$ transfer characteristics showed that both 1&3F-Inverter and 2&4F-Inverter properly operated with excellent switching behavior and a DC gain up to 35 V per V at $V_{DD} = 5\,V$ and $8\,V$ (Fig. 4d–i). Only negligible hysteresis ($\Delta V = \sim 0.18\,V$) was observed for both 1&3F- and 2&4F-Inverter. The measured $I_{DD}$ in both the inverters was $\sim 100\,pA$ level ($\sim 1 \times 10^{-10}\,A$ at $V_{IN} = 0$ V and $\sim 2 \times 10^{-12}\,A$ at $V_{IN} = 50\,V$), which is fully consistent with our repeated observations from DNTT-based and PTCDI-C13-based OTFTs with PV3D3 dielectric layer in that the off-current of the OTFT was as low as $\sim 100\,pA$[28] (Fig. 3f-j). To compare the proposed 3D-integrated inverters with conventional 2D inverters, we also fabricated DNTT and PTCDI-C13-based inverter with conventional inverter structure (Supplementary Fig. 20). The electrical characteristics of the proposed 3D inverters were comparable to the characteristics of the conventional 2D inverter, but the density of the 3D inverter circuit was significantly better (4 times higher). Additional investigation results for the inverter circuits, including transient measurement data (Supplementary Fig. 21) and air-stability of the 3D inverter circuit (Supplementary Fig. 13) are given.

We also demonstrated NAND and NOR logic gates by using the proposed vertical interconnect scheme. Two n-type transistors and two p-type transistors are stacked vertically, so that NAND and NOR logic circuits requiring four transistors can be implemented in a unit area (Supplementary Fig. 22a–d). Both the 3D-stacked NAND and NOR circuits properly operated according to the applied $V_A$ and $V_B$. The measured $I_{DD}$ in both the logic circuits was $\sim 500\,pA$. We also measured the transient logic operation by simultaneously applying the two input voltage pulses ($V_A$ and $V_B$). In the transient measurement, $V_A$ and $V_B$ are applied as a function of time and the switching characteristics of NAND and NOR gates showed that a corresponding output voltage changed properly according to logical truth table (Supplementary Fig. 23a, b)

## Discussion

In summary, we proposed a novel solvent-free multilevel metal interconnect strategy using patterned PV3D3 dielectric polymer. Rather than adopting a via-hole formation process involving local removal of the insulator, direct implementation of patterned PV3D3 dielectric polymer was used to selectively provide opened areas for vertical interconnections. The proposed scheme allows OTFT devices to be simply interconnected without degradation of organic semiconductors. By using the proposed method, we successfully demonstrated vertically stacked OTFTs on 5 layers with ideal transfer and output characteristics. We also demonstrated 3D-integrated inverters on 4 layers, which exhibited excellent inverter characteristics. To the best of our knowledge, this is the first demonstration of OTFTs with up to 5 stacking layers (Supplementary Table 1). The proposed multilevel interconnection scheme shares the design philosophy of the metal-interconnection scheme used in silicon-based ICs, i.e., the via-holes are simultaneously formed either with via-hole etching (silicon-based IC case) or a patterned dielectric layer (this work). As PV3D3 is compatible with photo-[42] and e-beam lithography processes[43] (Supplementary Fig. 24), the proposed via-hole-less interconnection scheme can provide the 3D-circuits with finer patterns if the semiconductor layers are changed to lithography-compatible materials[44–47]. Although further down-scaling of device size is beyond the scope of this work, our study provides a fundamental technology for metal interconnection to enable 3D vertical integration of a larger number of devices for a given area budget. As a robust and scalable metal interconnection is crucial for the success of silicon-based ICs, we believe that the results of this work are important for the development of very large-scale organic electronics that require complex metal routing.

## Methods

**Materials**. The V3D3 monomer (1,3,5-trimethyl-1,3,5-trivinyl cyclotrisiloxane, Gelest, 95%) and initiator TBPO (tert-butyl peroxide, Aldrich, 97%) for the polymer dielectric deposition process were purchased from commercial sources and used as received. The p- and n-type semiconductors, dinaphtho[2;3-b:2′,3′-f]-thieno[3,2-b]thiophene (DNTT) and N,N′-ditridecylperylene-3,4,9,10-tetra-carboxylic diimide (PTCDI-C13), respectively, were purchased from Sigma-Aldrich.

**Patterned PV3D3 film deposition**. V3D3 and TBPO were vaporized and delivered to a custom-built iCVD chamber (Supplementary Fig. 25). The flow rate ratio of V3D3 and TBPO was 2.5:1 and controlled by a needle valve. The process pressure was 300 mTorr, and the filament was heated to 130 °C. The bottom cool-stage temperature was maintained at 40 °C. The shadow mask (Invar) was tightly contacted with glass or $SiO_2/Si$ wafer by a magnet to maintain the alignment of the shadow mask and substrate and minimize the gap between them while the iCVD process was performed. The deposition rate of the PV3D3 layer on a magnetic plate was $0.625\,nm\,min^{-1}$. Further characterizations of the PV3D3 film pattern are provided in Supplementary Figs. 26–29.

**Deposition of organic semiconductor and metal layers**. Al gate electrodes and Au contact electrodes were deposited by thermal evaporation with deposition rates of $\sim 1.0\,Å\,s^{-1}$ and $\sim 0.5\,Å\,s^{-1}$, respectively. PTCDI-C13 and DNTT layers were deposited via thermal evaporation with a deposition rate of $\sim 0.3\,Å\,s^{-1}$. The chamber pressure was $< 10^{-6}$ Torr.

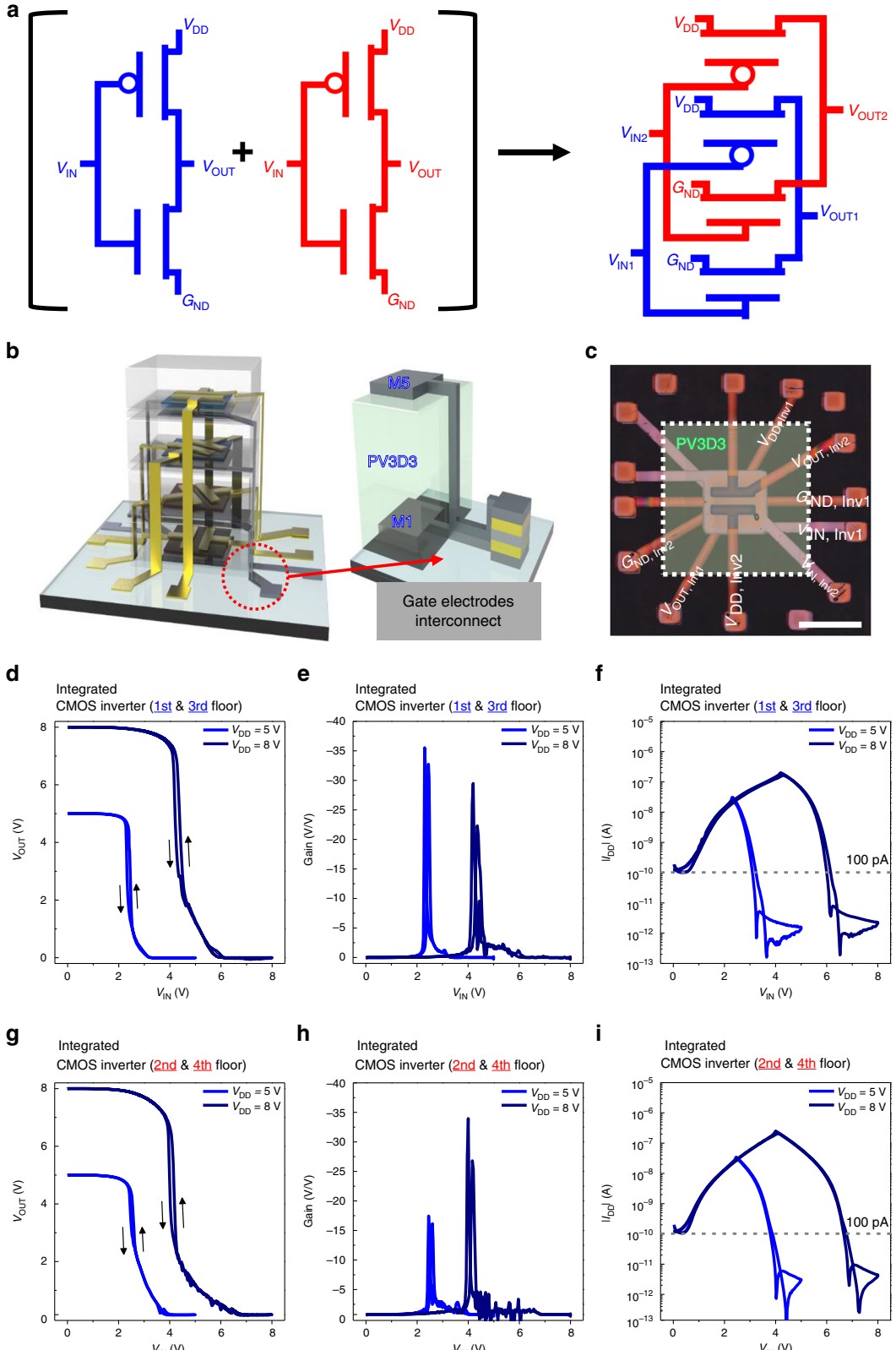

**Fig. 4** Two 3D-stacked inverter circuits. **a** Schematic of two 3D-stacked inverter circuits and a conventional inverter circuit. **b** A schematic of the gate metal routing line in two 3D-stacked inverter circuits. **c** An optical micrograph image of two 3D-stacked inverter circuits, scale bar, 1400 μm. **d**, **e**, **f** The inverter characteristics of a 3D-stacked inverter circuit integrated at 1F and 3F. Output voltage curve with respect to the input voltage value (**d**). DC gain plot with respect to the input voltage value (**e**). Leakage current plot with respect to the input voltage value (**f**). **g**, **h**, **i** Inverter characteristics of a 3D-stacked inverter circuit integrated at 2F and 4F. Output voltage curve with respect to the input voltage value (**g**). DC gain plot with respect to the input voltage value (**h**). Leakage current plot with respect to the input voltage value (**i**)

**Fabrication of the metal interconnect test MIM structure**. For the multimetal interconnect test structure, one Al bottom-electrode layer (line width = 200 μm) and four Au-electrode layers (line width = 200 μm) were used, and each electrode layer was isolated by patterned iCVD dielectric layers (thickness = 30 nm). The measured resistance value in a single Au-electrode line was ~ 0.23 kΩ. The measured value was calculated by normalizing the width/length to that of the multimetal interconnect test structure. Three types of metal-insulator-metal (MIM) test structures were fabricated via thermal evaporation and an iCVD chamber with shadow-mask patterning. To verify the lateral conduction test sample, top Au lines were deposited on 3 patterned dielectric structures with a 1 mm width, 1 mm pitch, and thickness varying in the range from 50 to 800 nm. Both test structures for the dielectric pattern margin over the bottom electrode and the distance from the metal interconnection used a patterned dielectric layer with a thickness of approximately 50 nm.

**Fabrication of 3D-stacked OTFTs and two 3D-stacked CMOS inverter**. The 25 × 25 mm$^2$ glass substrates were used and cleaned by ultrasonication with detergent dissolved in deionized (DI) water, acetone and isopropanol (IPA) for 20 min. The substrates were blown dry with N$_2$ gas. A 50-nm-thick Al gate electrode layer was deposited and followed by deposition of a patterned iCVD gate dielectric layer with a thickness of ~ 50 nm. A 30-nm-thick layer of DNTT was used as a p-type semiconductor, and a 30-nm-thick layer of PTCDI-C13 was used as an n-type semiconductor. Then, a 50-nm-thick Au source/drain electrode was deposited through a shadow mask with channel dimensions of 1000 μm (W) × 200 μm (L). In the case of the n-type TFTs, the PTCDI-C13 films were thermally annealed at 200 °C for 1 h prior to the contact electrode deposition. For 3D-stacked TFT devices, two n-type TFTs and three p-type TFTs were stacked sequentially, and each TFT was isolated by patterned PV3D3 isolating layers (thickness = 1 μm). All fabrication processes were conducted in a N$_2$-purged box.

**Electrical characterization of devices**. The electrical characteristics of all MIM and 3D-stacked devices and circuits were measured using a probe station and Keithley 4200-SCS instrument. All measurements were performed under ambient atmosphere.

## Data availability

The data that support the findings of this study are available from the corresponding authors upon reasonable request.

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

## Acknowledgements

This work was supported by the MSIP (Ministry of Science, ICT and Future Planning), Korea, under the "ICT Consilience Creative Program" (IITP-2018-2011-1-00783) supervised by the IITP (Institute for Information & communications Technology Promotion), the National Research Foundation of Korea (NRF) grant funded by the Korean government (MSIP) (No. NRF-2017R1A2B4006749), the National Research Foundation of Korea (NRF) grant funded by the Korean government (MEST) (NRF-2017R1A2B3007806), and the Samsung Research Funding Center of Samsung Electronics under Project Number SRFCMA1402-04.

## Author contributions

H.Y., H.P., H.S., S.G.I., and J.-J.K. conceived the idea and designed the experiments. H.Y. and H.P designed, fabricated, and measured all the devices and circuits. S.Y. helped with the circuit design and characterization of the patterned PV3D3 structures. H.Y. and O.S. performed scanning electron microscopy (SEM) and optical microscopy (OM) measurements of the fabricated devices. H.Y., H.P., S.G.I., and J.-J.K. wrote the manuscript. All authors reviewed the manuscript and discussed the results.

## Additional information

**Competing interests:** The authors declare no competing interests.

