## [Peer Review File · Nature Communications]

Reviewers' comments:

Reviewer #1 (Remarks to the Author):

This is relatively interesting manuscript. However, several points are unclear. Authors should add further informations and data for readers.

- 1) In the fabrication process, authors used the laser irradiation, iCVD process ad shadow mask. This is not suitable process for fine patterning and highly integration of circuits. Is there any idea to overcome this issue ?
- 2) In the inverter characteristics shown in Fig3f, the IDD currents are very high, about 10-8A. Why the IDD is high in the CMOS inverter ? This is not practical for power consumption.
- 3) Authors should draw the overall concept of the total fabrication process to for the readers.
- 4) Device stability in air is crucial for the practical applications. The data for long-term stability should be added.
- 5) Reproducibility in the device performances is also very important for the applications. The distribution of the performances of devices(at least 10 devices) should be shown in the manuscript.

Reviewer #2 (Remarks to the Author):

This manuscript report an approach to fabricate a multilevel metal interconnect scheme. 3D organic integrated circuits were constructed with solvent-free patterning of insulator layers to form an interconnecting area. Dielectric layers were deposited on selected locations through a shadow-mask pattern. Such a scheme guaranteed less via-hole within the circuits. However, this scheme employed shadow mask for the whole process which limit the shrink of the device size, thus maybe not suitable for the very large scale organic electronics. Thus, I am fairly inclined to reject the manuscript to be published on Nature Communications.

Manuscript #: NCOMMS-18-30118

Title: Highly Stacked, 3D Organic Integrated Circuits with Via-Hole-Less, Multilevel Metal Interconnects

Response to the Reviewers' Comments

The authors thank the Reviewers for their thorough reading of our manuscript and the useful comments. We have revised the manuscript and supporting information based on the Reviewers' comments. The Reviewers' comments appear in **black**, and the authors' responses in **blue**. In the revised manuscript, **the changes** made in this revision are highlighted in yellow.

Reviewer (#1)'s COMMENTS:

This is relatively interesting manuscript. However, several points are unclear. Authors should add further information and data for readers.

1) In the fabrication process, authors used the laser irradiation, iCVD process and shadow mask. This is not suitable process for fine patterning and highly integration of circuits. Is there any idea to overcome this issue?

Response:

We recognize the Reviewer's concern about the scalability of the fabrication processes used in this study. Both scaled S/D electrodes and finer iCVD dielectric layer patterns are readily obtainable via photo- and e-beam lithography processes. We have already demonstrated that the iCVD-based PV3D3 dielectric layer is compatible with both conventional photolithography^[43,R1] and e-beam lithography lift-off^[44,R2] to provide far smaller pattern of PV3D3 layer (~500 nm). The electrical properties of PV3D3 are maintained after exposure to the developer solvent (AZ developer, MicroChemicals) for 30 s, as shown in **Fig. R1a**. The 22 nm-thick PV3D3 layer maintained its high capacitance value ($C_{diel} = 89 \text{ nFcm}^{-2}$) and insulation performance with the breakdown field higher than 2 MV/cm even after dipping the ultrathin insulating layer in developer (**Fig. R1b-d**). The main limitations in scaling result from the instability of the organic semiconductors. Exposure to chemical solvents, heat, and UV irradiation during the patterning process for the dielectric layers or the electrodes degrades the device performance of the organic electronic devices. Additionally, damage-free, finer

patterning of most organic semiconductor with photolithography is quite challenging. While beyond the scope of this study, robust, lithography-compatible semiconductors such as IGZO^[45,R3], carbon nanotubes^[46,R4], transition metal dichalcogenides^[47,R5], or photolithography-compatible polymers, such as recently reported PTDPSe-SiC4^[48,R6], instead of damage-prone DNTT and PTCDI-C13, can be incorporated for the fabrication of 3D-stacked transistor circuits with finer patterns.

To clarify, we implemented ultrathin PV3D3 dielectric patterns *in situ* via shadow mask without using laser irradiation, which enabled via-hole-free interlayer metal connection to achieve a high-performance 3D stack of organic transistors and circuits with 100% device yield without compromising the device performance of each component device performance.

We revised the manuscript by adding the above discussion on this fine patterning aspect as follows;

- (1) Revised the main text in the revised manuscript (p. 15)
→ As PV3D3 is compatible with photo-^[43] and e-beam lithography processes^[44], the proposed via-hole-less interconnection ... lithography-compatible materials^[45-48].
- (2) Added the characteristics of PV3D3 dielectric layer before/after development in Supplementary Information (p. 32)
- (3) Added the reference [43-48] in the revised manuscript (p.23)

Figure R1. Characteristics of PV3D3 dielectric layer before/after development. **a**, Schematic illustration of dipping PV3D3 in developer solvent. **b**, C_{diel} - V characteristics of 22 nm PV3D3 dielectric layer between Al electrodes before/after development process. **c**, leakage current densities (J)-applied voltage (V) measured for 22 nm PV3D3 before/after development process. **d**, leakage current densities (J)-applied electric field (E_i) measured for 22 nm PV3D3 before/after development process.

[43,R1] Park, C. W., et al. "Stretchable active matrix of oxide thin-film transistors with monolithic liquid metal interconnects." *Applied Physics Express* **11**, 126501 (2018).

[44,R2] Oh, J. G., et al. "A High-Performance Top-Gated Graphene Field-Effect Transistor with Excellent Flexibility Enabled by an iCVD Copolymer Gate Dielectric." *Small* **14**, 1703035 (2018).

[45,R3] Heremans, P., et al. "Flexible metal-oxide thin film transistor circuits for RFID and health patches." *2016 IEEE International Electron Devices Meeting (IEDM)*. IEEE, 2016.

[46,R4] Sangwan, V. and Beck, M. et al. "Self-Aligned van der Waals Heterojunction Diodes and Transistors." *Nano letters* **18**, 1421-1427 (2018).

[47,R5] Kim, S., et al. "High-mobility and low-power thin-film transistors based on multilayer MoS₂ crystals." *Nat. Commun.* **3**, 1011 (2012).

[48,R6] Lee, E., et al. "Chemically robust ambipolar organic transistor array directly patterned by photolithography." *Adv. Mater.* **29**, 1605282 (2017).

2) In the inverter characteristics shown in **Fig. 3f**, the I_{DD} currents are very high, about 10^{-8} A. Why the I_{DD} is high in the CMOS inverter? This is not practical for power consumption.

Response:

Thank you for the Reviewer's considerate comment. We rechecked the I_{DD} - V_{IN} curve data for **Fig. 3f**, and found that the source measurement unit (SMU) with medium current range was used to measure the I_{DD} when we measured the 3D-stacked inverter circuits. We reproduced the two 3D-stacked inverter circuits and measured the inverter characteristics with lower current range SMU again (**Fig. R2**). Both 1&3F-Inverter and 2&4F-Inverter properly operated with good switching behavior. The measured I_{DD} in both the inverters was ~ 100 pA level ($\sim 1 \cdot 10^{-10}$ A at $V_{IN} = 0$ V and $\sim 2 \cdot 10^{-12}$ A at $V_{IN} = 50$ V), which is fully consistent with our repeated observations from DNTT-based and PTCDI-C13-based OTFTs with PV3D3 dielectric layer in that the off-current of the OTFT was as low as ~ 100 pA^[29] (**Fig. 3f** and **Fig. R12**).

To further investigate the I_{DD} characteristics in the proposed 3D-stacked complementary circuits, we fabricated NAND and NOR logic gates by using the proposed vertical interconnect scheme (**Fig. R3a-d**). Both the 3D-stacked NAND and NOR circuits properly operated in accordance with the applied V_A and V_B . The measured I_{DD} in both the logic circuits was ~ 500 pA. We also measured the transient logic operation by simultaneously applying the two input voltage pulses (V_A and V_B). In the transient measurement, V_A and V_B are applied as a function of time and the resultant switching characteristics of NAND and NOR gates showed that a corresponding output voltage changed properly according to logical truth table (**Fig. R4a,b**).

We revised the manuscript by adding the above discussion on this I_{DD} aspect as follows;

(1) Revised the main text and added the reproduced 3D-intergrated inverters results in the revised manuscript (p.13-14)

→ The V_{OUT} - V_{IN} transfer characteristics showed that both 1&3F-Inverter and 2&4F-Inverter properly operated ... the off-current of the OTFT was as low as $\sim 100 \text{ pA}^{29}$ (Fig. 3f).

(2) Replaced with the remeasured 3D-stacked inverter circuit results (Fig. 4d-i) in the revised manuscript (p. 28-29)

(3) Added the 3D-integrated NAND and NOR results in the revised manuscript (p. 14)
 → We also demonstrated NAND and NOR logic gates by using the proposed vertical interconnect scheme... according to logical truth table (Supplementary Fig. 29a,b)

Figure R2. Reproduction result of two 3D-stacked inverter circuits. a, Symbol of the inverter circuit. **b,** Schematic of two 3D-stacked inverter circuits. **c,** DC transfer characteristics of a 3D-stacked inverter circuit integrated on 1F and 3F. **d,** The corresponding I_{DD} plot with

respect to the input voltage value. **e**, DC transfer characteristics of a 3D-stacked inverter circuit integrated on 2F and 4F. **f**, The corresponding I_{DD} plot with respect to the input voltage value.

Figure R3. 3D-stacked NAND and NOR logic circuits. a, Symbol of NAND circuit. **b**, Symbol of NOR circuit. **c**, Schematic of 3D-stacked NAND circuit. **d**, Schematic of 3D-stacked NOR circuit. **e**, DC transfer characteristics of NAND circuit. **f**, The corresponding I_{DD} plot of NAND circuit. **g**, DC transfer characteristics of NOR circuit. **h**, The corresponding I_{DD} plot of NOR circuit.

Figure R4. Transient measurement of 3D-stacked NAND and NOR logic circuits. a, Measured transient result of NAND circuit with respect to V_A and V_B . **b,** Measured transient result of NOR circuit with respect to V_A and V_B .

3) Authors should draw the overall concept of the total fabrication process to for the readers.

Response:

We thank the Reviewer for the constructive suggestion. To illustrate the detailed fabrication process more clearly, we redraw the overall concept of the entire process for the 3D-stacked OTFTs and inverter circuits, as shown in **Fig. R5** and **R6**. The total fabrication process consists of five transistor manufacturing process (1F-5F). Each transistor manufacturing process consists of five deposition processes: (1) Al gate electrode, (2) PV3D3 dielectric layer for gate dielectric layer, (3) semiconductor layer, (4) Au contact electrode, and (5) PV3D3 dielectric layer for intermetal dielectric layer (**Fig. R5**). The whole fabrication process results in a five-transistor 3D stack (**Fig. R6**).

Figure R5. Schematic of fabrication process flow for the 3D-stacked OTFTs.

Figure R6. Schematic and optical images of transistors are shown as transistors are stacked up. Scale bar, 1,400 μm

For the 3D-stacked inverter circuits, two complementary inverter circuits were fabricated in stacked manner, consisting of four OTFTs: two n-type OTFTs on 1F-2F and two p-type OTFTs on 3F-4F (**Fig. R7**). The OTFT was surrounded by an open area through which the electrodes were connected vertically to each other. **Fig. R8** shows an example of the metal interconnection of the gate electrode on 1F and the other gate electrode on 3F. The gate

electrodes on 2F and 4F, the drain electrodes on 1F and 3F, and the drain electrodes on 2F and 4F were interconnected respectively, in the same manner (Fig. R9).

Figure R7. Schematic of fabrication process flow for the 3D-stacked inverter circuits.

Figure R8. Schematic of the gate metal interconnection in the 3D-stacked inverters. Scale bar, 1,400 μm

Figure R9. Schematic and optical images of the 3D-stacked inverter circuits are shown as transistors are stacked up. Scale bar, 1,400 μm

Based on the above discussion, we have revised our manuscript as follows;

- (1) Revised the main text in the revised manuscript (p.10)
 → A more detailed manufacturing process and ... OTFTs is shown in Supplementary Fig. 11-12.
- (2) Revised the main text in the revised manuscript (p.13)
 → A more detailed manufacturing process and ... 3D-stacked inverter circuits is shown in Supplementary Fig. 22-24.
- (3) Added the description of the 3D-stacked inverter circuits in Supplementary Information (p.27)
 → For the 3D-stacked inverter circuits, two complementary ... the drain electrodes on 2F and 4F were interconnected respectively, in the same manner.

4) Device stability in air is crucial for the practical applications. The data for long-term stability should be added.

Response:

As the Reviewer suggested, we reproduced the 3D-stacked transistors and investigated their environmental stability. To implement more stable device and circuit in air ambient, we

encapsulated the 3D-stacked organic transistors with 10 nm-thick Al₂O₃ layer via atomic layer deposition (ALD) process (**Fig. R10a**). Both n-type and p-type 3D-stacked OTFTs were measured at in an inert N₂ atmosphere first, and then stored and measured in air ambient (20 °C, 45% relative humidity). To investigate the effect of the ultrathin Al₂O₃ encapsulation on the device stability, the 3D-stacked devices without the encapsulation layer were also characterized as the negative control.

For the Al₂O₃-encapsulated 3D-stacked OTFTs, the initial device characteristics were fully maintained throughout the entire measurement period of ~214 h (**Fig. R10b**). For the devices without encapsulation, on the other hand, the PTCDI-C13-based n-type transistors showed substantial mobility decrease and V_{TH} shift with the increased air exposure time (**Fig. R10c**). The DNNT-based p-type transistors also showed quite huge V_{TH} shift along with the increased air exposure time (**Fig. R10d**), which is ascribed to the oxygen and moisture in previous reports^[41, R7, 42, R8]. In the Al₂O₃-encapsulated 3D-stacked OTFTs, the 10 nm Al₂O₃ layer was sufficiently good for the protection of the 3D stack devices from the exposure of oxygen and moisture, leading to the high air-stability result.

To further evaluate the air-stability of the Al₂O₃-encapsulated 3D-stack OTFTs, we also measured the air stability of the Al₂O₃-encapsulated 3D-stacked inverter as a function of the air exposure time (**Fig. R11a,b**), where the voltage transfer curve of the inverter circuit remained practically identical, demonstrating its excellent environmental stability.

Thank you for the Reviewer's constructive suggestion. We revised the manuscript by adding the above discussion on the air-stability aspect as follows;

- (1) Revised the main text in the revised manuscript (p. 11-12)
→ We also investigated the air-stability of the 3D-OTFTs. To implement more stable device and circuit in air ambient... its excellent environmental stability.
- (2) Revised the main text in the revised manuscript (p. 14)
→ Additional investigation results for the inverter circuits ... and air-stability of the 3D inverter circuit (Supplementary Fig. 19) are given.
- (3) Added the Al₂O₃ encapsulated 3D-stacked transistors and inverter circuit results in Supplementary Information (p. 20-21)
- (4) Added the reference [41-42] in the revised manuscript (p.23)

[41,R7] Roh, J., et al. "Air stability of PTCDI-C13-based n-OFETs on polymer interfacial layers." *Phys. Status Solidi RRL* **7**, 469-472 (2013).

[42,R8] Di Pietro, R., et al. "Spectroscopic investigation of oxygen-and water-induced electron trapping and charge transport instabilities in n-type polymer semiconductors." *J. Am. Chem. Soc.* **134**, 14877-14889 (2012).

Figure R10. Air-stability of 3D-stacked OTFTs. a, Schematic structures of the Al_2O_3 encapsulated 3D-stacked OTFTs. b, Plot of mobility and V_{TH} as a function of the air-exposure time for the n-type 3D-stacked OTFT. c, Plot of mobility and V_{TH} as a function of the air-exposure time for the p-type 3D-stacked OTFT. d, Transfer curves of p-type and n-type 3D-stacked OTFTs with Al_2O_3 encapsulation.

Figure R11. Air-stability of the Al_2O_3 encapsulated 3D-stacked inverter circuits. a, Inverter transfer characteristics as a function of the air-exposure time for the Al_2O_3 encapsulated 3D-stacked inverter circuits. **b,** Plot of pull-up and pull-down V_{OUT} values as a function of the air-exposure time.

5) Reproducibility in the device performances is also very important for the applications. The distribution of the performances of devices (at least 10 devices) should be shown in the manuscript.

Response:

As the Reviewer suggested, we characterized 20 devices to investigate the yield, uniformity, and reproducibility of the 3D stack devices. In this study, a glass substrate (2.5 cm by 2.5 cm) includes four 3D-stacked OTFT buildings and one building consisting of 3 n-type OTFTs and 2 p-type OTFTs (**Fig. R12**). A total of 20 OTFTs were measured for statistical investigation. To verify the switching operation in both linear and saturation regions, the transistors were measured at $V_D = 2$ V and 10 V, respectively (**Fig. R13**). All OTFTs exhibited transistor operation with excellent device performance including high on/off current ratio $\sim 10^5$ and 100% yield. Furthermore, no I - V hysteresis was observed in all the transfer characteristics.

For PTCDI-C13-based n-OTFTs, the average saturation and linear mobility values were $\mu_{sat} = 0.56 \pm 0.048 \text{ cm}^2\text{V}^{-1}\text{s}^{-1}$ and $\mu_{lin} = 0.50 \pm 0.054 \text{ cm}^2\text{V}^{-1}\text{s}^{-1}$, respectively (**Fig. R14**). For DNTT-based p-OTFTs, the average saturation and linear mobility values were $\mu_{sat} = 0.55 \pm 0.077 \text{ cm}^2\text{V}^{-1}\text{s}^{-1}$ and $\mu_{lin} = 0.27 \pm 0.055 \text{ cm}^2\text{V}^{-1}\text{s}^{-1}$, respectively.

We revised the manuscript by adding the above discussion on the reproducibility aspect as follows;

(1) Revised the main text and added the reproduced 3D-intergrated inverters results in the revised manuscript (p. 12)

→ To study the reproducibility and uniformity of the proposed 3D-OTFTs, we reproduced four 3D stacked OTFT buildings with 20 devices..., the average saturation and linear mobility values were $\mu_{sat} = 0.55 \pm 0.077 \text{ cm}^2\text{V}^{-1}\text{s}^{-1}$ and $\mu_{lin} = 0.27 \pm 0.055 \text{ cm}^2\text{V}^{-1}\text{s}^{-1}$, respectively.

(2) Added the reproducibility and uniformity results in Supplementary Information (p. 21-22)

Figure R12. A photograph of the fabricated die including 5 metal layers test structure, 3D-OTFTs, 3D-inverters, and 3D-stacked NAND and NOR circuits.

Figure R13. Transfer characteristics of 20 OTFTs (12 n-OTFTs and 8 p-OTFTs).

Figure R14. Reproducibility and uniformity for 3D-stacked OTFTs. a, Histograms of carrier mobility of 20 OTFTs under saturation region ($V_D = 10$ V). **b,** Histograms of carrier mobility of 20 OTFTs under linear region ($V_D = 2$ V). **c,** Histograms of on/off current ratio of 20 OTFTs.

Reviewer (#2)'s COMMENTS:

This manuscript report an approach to fabricate a multilevel metal interconnect scheme. 3D organic integrated circuits were constructed with solvent-free patterning of insulator layers to form an interconnecting area. Dielectric layers were deposited on selected locations through a shadow-mask pattern. Such a scheme guaranteed less via-hole within the circuits. However, this scheme employed shadow mask for the whole process which limit the shrink of the device size, thus maybe not suitable for the very large scale organic electronics. Thus, I am fairly inclined to reject the manuscript to be published on Nature Communications.

Response:

We fully agree with the Reviewer that the use of the shadow mask cannot provide finer patterning than the conventional etching-based lithography. It would be also highly desirable to obtain VLSI organic electronics with far higher circuit density by shrinking down the device size. However, we believe that there is a common consensus in this field that the value of organic electronic device and integrated circuits therefrom does not just originate from high density integration of organic electronic devices, but the unique advantageous characteristics of organic electronic devices such as mechanical flexibility^[R16], compatibility with various form factors^[R17,R18], huge accessibility to chemical synthesis^[R19], mass scalability^[R20] in fabrication process represented as the printable electronics, and cost competitiveness. For last three decades, many researchers had strived for the realization of various kinds of electronic device products with unprecedented product features such as wearable sensors^[R21], disposable electronics^[R22], foldable displays^[R22, R23] and so forth, by harnessing such desirable characteristics of organic electronics. Such ground-breaking achievements clearly illustrate that the down-scalability of organic electronic device products may be one of the important issues to be considered, but it cannot be the only decision-making factor in this field.

There have been a variety of attempts to achieve patterning of organic electronic materials without sacrificing their electronic performances. The biggest hurdle for high density pattern of organic electronic devices is the extremely high susceptibility of organic materials to environmental stress including chemical solvents, heat, and UV irradiation, which are all very frequently utilized to achieve high-fidelity pattern for conventional semiconductor industry^[48,R6, R24]. While beyond the scope of this work, it is imperative to develop alternative

patterning methods for organic electronic materials that do not damage the organic materials. For this purpose, additive patterning such as inkjet printing, reverse offset printing, gravure printing or shadow mask patterning have been investigated intensively to define fine channel dimension on organic semiconductors with minimal damage (**Fig. R15**).

Figure R15. Comparison of the channel length used in organic circuits according to the patterning process.

For the development of integrated 3D stack of OTFT arrays, securing a method to deposit a subsequent pattern without damaging the layer underneath is a non-trivial challenge. Using solution-based printing technology involves repeated exposure of the organic layers to a solvent, which risks serious damage through diffusion into the organic layer stack. Additionally, the lack of patternable, ultrathin, but high-performing organic dielectric layers has impaired the development of integrated circuits from organic electronic devices. Most gate dielectric materials that are compatible with printing processes lose their insulating properties below 200 nm thicknesses^[R25,R26], making the via-hole fabrication process extremely difficult. As devices are vertically stacked, the total thickness of the intermetal dielectric layers also increases, and thus forming via-holes through the layers becomes difficult. Despite significant effort, the number of multi-metal layers in 3D integrated organic transistors has been limited to 4 with the number of vertically stacked transistors no higher than 2, as shown in **Table R1**.

To achieve highly stacked 3D-OTFTs beyond the current capability, we present in this manuscript a new fabrication process for 3D-stack OTFT integration, whose characteristic features include (1) a simple vertical interconnect scheme to support a large number of vertical layers and targeted interconnects, (2) a low process temperature and (3) a robust dielectric deposition step that does not affect previously stacked devices. In this study, an ultrathin, but highly insulating organic gate dielectric layer was patterned in situ via shadow mask, which enabled via-hole-free interlayer metal connection to achieve a high-performance 3D stack of organic transistors and circuits with 100% device yield without compromising the device performance. Using this methodology, we demonstrated the ultrahigh stacking of OTFTs up to 5 transistors and 25 layers, the highest stacked organic transistors to date. In theory, we can stack well beyond 25 layers due to the versatility of the patterning method. Furthermore, we integrated two 3D-stacked complementary inverter circuits using transistors on 4 different floors, which, to the best of our knowledge, is most layers that have been used.

Table R1. Comparison of 3D-stacked transistors in terms of the number of stacked transistors, metal layers, and dielectric layers and the interconnect method.

	Via-hole-less dielectric pattern method	Interconnect method not reported			Laser-drilling method	Inkjet-printing method
	This work	[12]	[13]	[17]	[15]	[16]
# of dielectric layers	~9	~3	~2	~2	~4	~3
# of stacked transistors	5	2	2	2	2	2
# of metal layers	20	3	3	3	~4	3
Operating voltage (V)	~10	~6	~8	~30	~20	~30

If required, both scaled S/D electrodes and finer iCVD dielectric layer patterns are readily obtainable via photo- and e-beam lithography processes. We have already demonstrated that the iCVD-based PV3D3 dielectric layer is compatible with both conventional photolithography^[43,R1] and e-beam lithography lift-off^[44,R2] to provide far smaller pattern of

PV3D3 layer (~500 nm). The electrical properties of PV3D3 are maintained after exposure to the developer solvent (AZ developer, MicroChemicals) for 30 s, as shown in **Fig. R1a**. The 22 nm-thick PV3D3 layer maintained its high capacitance value ($C_{diel} = 89 \text{ nFcm}^{-2}$) and insulation performance with the breakdown field higher than 2 MV/cm even after dipping the ultrathin insulating layer in developer (**Fig. R1b-d**). The main limitations in scaling result from the instability of the organic semiconductors. Exposure to chemical solvents, heat, and UV irradiation during the patterning process for the dielectric layers or the electrodes degrades the device performance of the organic electronic devices. Additionally, damage-free, finer patterning of most organic semiconductor with photolithography is quite challenging. While beyond the scope of this study, robust, lithography-compatible semiconductors such as IGZO^[45,R3], carbon nanotubes^[46,R4], transition metal dichalcogenides^[47,R5], or photolithography-compatible polymers, such as recently reported PTDPSe-SiC4^[48,R6], instead of damage-prone DNTT and PTCDI-C13, can be incorporated for the fabrication of 3D-stacked transistor circuits with finer patterns.

Based on the discussion above, we anticipate that the fabrication methodology for 3D-stacked organic transistor circuits developed in this study will translate beyond academic endeavors into further integrated organic circuitry for future flexible electronics industry. Although further down-scaling of device size is beyond the scope of this work, our study provides a fundamental technology for metal interconnection to enable 3D vertical integration of a larger number of devices for a given area budget. Considering the tremendous impact of the 3D stacking of organic integrated circuits, with high performance and patternable gate insulators, especially those for flexible/wearable applications with low power consumption, we strongly believe the present work is eligible for *Nature Communications*.

We revised the manuscript by adding the above discussion on this fine patterning aspect as follows;

- (1) Revised the main text in the revised manuscript (p. 15)
→ As PV3D3 is compatible with photo-^[43] and e-beam lithography processes^[44], the proposed via-hole-less interconnection ... lithography-compatible materials^[45-48].
- (2) Added the characteristics of PV3D3 dielectric layer before/after development in Supplementary Information (p. 32)

(3) Added the reference [43-48] in the revised manuscript (p.23)

(4) Revised the main text in the revised manuscript (p. 15-16)

→ Although further down-scaling of ... the development of very large-scale organic electronics that require complex metal routing.

Figure R1. Characteristics of PV3D3 dielectric layer before/after development. a, Schematic illustration of dipping PV3D3 in developer solvent. **b,** C_{diel} - V characteristics of 22 nm PV3D3 dielectric layer between Al electrodes before/after development process. **c,** leakage current densities (J)-applied voltage (V) measured for 22 nm PV3D3 before/after development process. **d,** leakage current densities (J)-applied electric field (E) measured for 22 nm PV3D3 before/after development process.

[43,R1] Park, C. W., et al. "Stretchable active matrix of oxide thin-film transistors with monolithic liquid metal interconnects." *Appl. Phys. Express* **11**, 126501 (2018).

[44,R2] Oh, J. G., et al. "A High-Performance Top-Gated Graphene Field-Effect Transistor with Excellent Flexibility Enabled by an iCVD Copolymer Gate Dielectric." *Small* **14**, 1703035 (2018).

[45,R3] Heremans, P., et al. "Flexible metal-oxide thin film transistor circuits for RFID and health patches." *2016 IEEE International Electron Devices Meeting (IEDM)*. IEEE, 2016.

[46,R4] Sangwan, V. and Beck, M. et al. "Self-Aligned van der Waals Heterojunction Diodes and Transistors." *Nano letters* **18**, 1421-1427 (2018).

[47,R5] Kim, S., et al. "High-mobility and low-power thin-film transistors based on multilayer MoS₂ crystals." *Nat. Commun.* **3**, 1011 (2012).

[48,R6] Lee, E., et al. "Chemically robust ambipolar organic transistor array directly patterned by photolithography." *Adv. Mater.* **29**, 1605282 (2017).

[R9] Z., Ute, et al. "Megahertz operation of flexible low-voltage organic thin-film transistors." *Org. Electron.* **14**, 1516-1520 (2013).

[R10] B., James W., et al. "Small contact resistance and high-frequency operation of flexible low-voltage inverted coplanar organic transistors." *Nat. Commun.* **10**, 1119 (2019).

[R11] Xu, W., et al. "Flexible all-organic, all-solution processed thin film transistor array with ultrashort channel." *Sci. Rep.* **6**, 29055 (2016).

[R12] Shiwaku, R. et al. Printed Organic Inverter Circuits with Ultralow Operating Voltages. *Adv. Electron. Mater.* **3**, 1600557 (2017).

[R13] Takeda, Y. et al. Organic Complementary Inverter Circuits Fabricated with Reverse Offset Printing. *Adv. Electron. Mater.* **4**, 5–9 (2018).

[R14] Münzenrieder, N., et al. "Flexible self-aligned amorphous InGaZnO thin-film transistors with submicrometer channel length and a transit frequency of 135 MHz." *IEEE Transactions on Electron Devices* **60**, 2815-2820 (2013).

[R15] Chen, Z., et al. "Graphene nano-ribbon electronics." *Physica E Low Dimens Syst Nanostruct.* **40**, 228-232 (2007).

[R16] Myny, K. The development of flexible integrated circuits based on thin-film transistors. *Nat. Electron.* **1**, 30-39 (2018).

[R17] Someya, T., Bao, Z. & Malliaras, G. G. The rise of plastic bioelectronics. *Nature* **540**, 379–385 (2016).

[R18] Wu, X. et al. Thermally stable, biocompatible, and flexible organic field effect transistors and their application in temperature sensing arrays for artificial skin. *Adv. Funct. Mater.* **25**, 2138–2146 (2015).

[R19] Quinn, J. T. E. E., Zhu, J., Li, X., Wang, J. & Li, Y. Recent progress in the development of n-type organic semiconductors for organic field effect transistors. *J. Mater. Chem. C* **5**, 8654–8681 (2017).

[R20] Park, H. et al. Proving Scalability of an Organic Semiconductor To Print a TFT-Active Matrix Using a Roll-to-Roll Gravure. *ACS Omega* **2**, 5766–5774 (2017).

[R21] Heo, J. S., Eom, J., Kim, Y. H. & Park, S. K. Recent Progress of Textile-Based Wearable Electronics: A Comprehensive Review of Materials, Devices, and Applications. *Small* **14**, 1–16 (2018).

[R22] Jeong, H. et al. Novel Eco-Friendly Starch Paper for Use in Flexible, Transparent, and Disposable Organic Electronics. *Adv. Funct. Mater.* **28**, 1–9 (2018).

[R23] Noda, M. et al. An OTFT-driven rollable OLED display. *J. Soc. Inf. Disp.* **19**, 316 (2011).

[R24] Jang, J. et al. The development of fluorous photolithographic materials and their applications to achieve flexible organic electronic devices. *Flex. Print. Electron.* **1**, (2016).

[R25] Tang, W. et al. Highly Efficient All-Solution-Processed Low-Voltage Organic Transistor with a Micrometer-Thick Low-k Polymer Gate Dielectric Layer. *Adv. Electron. Mater.* **2**, 1500454 (2016).

[R26] Ji, D. *et al.* Copolymer dielectrics with balanced chain-packing density and surface polarity for high-performance flexible organic electronics. *Nat. Commun.* **9**, 1–9 (2018).

We thank the Reviewers for their valuable comments, which greatly helped us to improve this manuscript.

Reviewer #1 (Remarks to the Author):

The manuscript is well revised and proper to be accepted by this journal.

Reviewer #2 (Remarks to the Author):

The authors have convinced me that the scheme employed shadow mask may also be suitable for the integration of organic electronics. And the revised manuscript have added more information about the long-term stability, reproducibility of the devices and also the detail of the fabrication process etc. It seems good enough to be accepted.

Reviewer #1 (Remarks to the Author):

The manuscript is well revised and proper to be accepted by this journal.

Response:

Thank you.

Reviewer #2 (Remarks to the Author):

The authors have convinced me that the scheme employed shadow mask may also be suitable for the integration of organic electronics. And the revised manuscript have added more information about the long-term stability, reproducibility of the devices and also the detail of the fabrication process etc. It seems good enough to be accepted.

Response:

Thank you.